

# Photosynthetic performance and stevioside concentration are improved by the arbuscular mycorrhizal symbiosis in *Stevia rebaudiana* under different phosphate concentrations

Luis G. Sarmiento-López[1], Melina López-Meyer[2], Gabriela Sepúlveda-Jiménez[1], Luis Cárdenas[3] and Mario Rodríguez-Monroy[1]

[1] Departamento de Biotecnología, Centro de Desarrollo de Productos Bióticos-Instituto Politécnico Nacional, Yautepec, Morelos, México
[2] Departamento de Biotecnología Agrícola, Centro Interdisciplinario de Investigación para el Desarrollo Integral Regional Unidad Sinaloa-Instituto Politécnico Nacional, Guasave, Sinaloa, México
[3] Departamento de Biología Molecular de Plantas, Instituto de Biotecnología-Universidad Nacional Autónoma de México, Cuernavaca, Morelos, México

Corresponding author
Mario Rodríguez-Monroy, mrmonroy@ipn.mx

## ABSTRACT

In plants, phosphorus (P) uptake occurs via arbuscular mycorrhizal (AM) symbiosis and through plant roots. The phosphate concentration is known to affect colonization by AM fungi, and the effect depends on the plant species. *Stevia rebaudiana* plants are valuable sources of sweetener compounds called steviol glycosides (SGs), and the principal components of SGs are stevioside and rebaudioside A. However, a detailed analysis describing the effect of the phosphate concentration on the colonization of AM fungi in the roots and the relationship of these factors to the accumulation of SGs and photochemical performance has not been performed; such an analysis was the aim of this study. The results indicated that low phosphate concentrations (20 and 200 $\mu$M KH$_2$PO$_4$) induced a high percentage of colonization by *Rhizophagus irregularis* in the roots of *S. rebaudiana*, while high phosphate concentrations (500 and 1,000 $\mu$M KH$_2$PO$_4$) reduced colonization. The morphology of the colonization structure is a typical *Arum*-type mycorrhiza, and a mycorrhiza-specific phosphate transporter was identified. Colonization with low phosphate concentrations improved plant growth, chlorophyll and carotenoid concentration, and photochemical performance. The transcription of the genes that encode kaurene oxidase and glucosyltransferase (*UGT74G1*) was upregulated in colonized plants at 200 $\mu$M KH$_2$PO$_4$, which was consistent with the observed patterns of stevioside accumulation. In contrast, at 200 $\mu$M KH$_2$PO$_4$, the transcription of *UGT76G1* and the accumulation of rebaudioside A were higher in noncolonized plants than in colonized plants. These results indicate that a low phosphate concentration improves mycorrhizal colonization and modulates the stevioside and rebaudioside A concentration by regulating the transcription of the genes that encode kaurene oxidase and glucosyltransferases, which are involved in stevioside and rebaudioside A synthesis in *S. rebaudiana*.

## INTRODUCTION

*Stevia rebaudiana* Bertoni is a plant that belongs to the Asteraceae family and accumulates compounds in its leaves called steviol glycosides (SGs) (*Brandle, Starratt & Gijzen, 1998*). Stevioside and rebaudioside A are the best-known SGs and are important compounds for human health because they are natural low-calorie sweeteners. The sweetening power of stevioside and rebaudioside A is 143 and 320 times higher than that of sucrose, respectively (*Lemus-Mondaca et al., 2012*; *Brandle & Telmer, 2007*). The biosynthetic pathway for SG synthesis begins in the chloroplasts with the synthesis of geranylgeranyl diphosphate (GGDP) generated from the MEP (methyl-erythritol-4-phosphate) pathway (*Totté et al., 2000*). GGDP is transformed to kaurene by two cyclization steps carried out by a terpene cyclase (*Brandle & Telmer, 2007*). In the endoplasmic reticulum, kaurene is oxidized by kaurene oxidase (KO) to kaurenoic acid; the oxidation of kaurenoic acid produces gibberellins, while the hydroxylation of kaurenoic acid produces steviol. The hydroxyl groups of steviol are glycosylated by the enzymes uridine diphosphate (UDP)-glycosyltransferases (UGTs), and the number of sugars attached by UGTs generates the various SGs (*Brandle & Telmer, 2007*). The UGT74G1 enzyme is involved in the conversion of steviolbioside to stevioside, while the UGT76G1 enzyme converts stevioside into rebaudioside A (*Kim et al., 2019*).

Arbuscular mycorrhizal (AM) symbiosis, between the phylum Glomeromycota and plants, is a mutualistic association that is useful in the culture of plants and has agricultural and medicinal importance. This symbiosis improves plant growth, photosynthesis and nutrient uptake and increases the production of phytochemicals (*Smith & Read, 2008*; *Spatafora et al., 2016*; *Schoefs et al., 2015*).

In plants, there are two modes of P uptake: one mode is by the plant's own Pi transporters, and the other mode occurs via AM symbiosis with mycorrhiza-specific phosphate transporters and takes place in the arbuscules (*Harrison, Dewbre & Liu, 2002*). Phosphate transporters are considered a key feature of this mycorrhizal symbiosis (*Karandashov & Bucher, 2005*). Mycorrhiza-specific phosphate transporters are expressed in arbuscule-containing cortical root cells and are thus considered general markers for AM symbiosis in different model plants (*Harrison, Dewbre & Liu, 2002*; *Nagy et al., 2005*; *Nouri et al., 2014*). Likewise, the phosphate concentration affects the colonization of roots by AM fungi Phosphate application at a high concentration may inhibit the formation of arbuscular mycorrhizae, and the sensitivity to phosphate and the grade of inhibition of arbuscule formation depend on the plant species (*Smith, Smith & Jakobsen, 2004*); for example, these factors differ in *Medicago truncatula* (*Bonneau et al., 2013*) and *Petunia hybrida* (*Breuillin et al., 2010*; *Nouri et al., 2014*).

In *S. rebaudiana*, AM symbiosis enhances the production of stevioside and rebaudioside A and involves nutritional and nonnutritional mechanisms (*Mandal et al., 2013*). AM

symbiosis also upregulates the transcription of eleven SG biosynthesis genes as a consequence of the improved nutrition status and the increase in photosynthesis in the plant (*Mandal et al., 2015*). This result suggests the roles of phosphorus nutrition and AM symbiosis in influencing SG concentration; fertilization with 25 mg $P_2O_5$ kg$^{-1}$ soil in association with AM symbiosis improved SG yield, P uptake and P nutrient use efficiency (*Tavarini et al., 2018*). However, a detailed analysis and systematic description of the morphological type of AM symbiosis and the effect of the different phosphate concentrations on the establishment of AM symbiosis, the identification of mycorrhiza-specific phosphate transporters, the photosynthetic performance, and the relationship with the accumulation of SGs in *S. rebaudiana* plants have not been addressed. Therefore, in this study, we reported the effects of different phosphate concentrations on the establishment of AM symbiosis between *Rhizophagus irregularis* and *S. rebaudiana*, their relationship with photochemical performance and the accumulation of steviol glycosides (SGs) and the expression of two key genes, *UGT74G1* and *UGT76G1*, which encode the (UDP)-glycosyltransferases involved in stevioside and rebaudioside A biosynthesis, respectively. The participation of a mycorrhiza-specific phosphate transporter as a key feature of this mycorrhizal symbiosis was demonstrated.

## MATERIALS & METHODS

### Inoculation with *Rhizophagus irregularis* and plant growth conditions

*R. irregularis* was provided by Dr. Melina López-Meyer from "Centro Interdisciplinario de Investigación para el Desarrollo Integral Regional, Unidad Sinaloa", Sinaloa state, Mexico. The inoculum was grown on Petri dishes with two compartments containing transformed carrot roots on minimum medium with 2% Gel-rite (Sigma-Aldrich) and incubated in the dark at $23 \pm 2$ °C for six months according to the method reported by *Bécard & Fortin (1988)*.

Six-month-old cuttings of *S. rebaudiana* plants were cultured under greenhouse conditions. Briefly, seven-cm cuttings were disinfected with 70% ethanol for 1 min and 2% sodium hypochlorite for 1 min and washed three times with sterile distilled water for 2 min. To measure root development, the cuttings were transplanted under hydroponic conditions in glass tubes containing Fahraeus medium and cultured in a controlled environment chamber at 25 °C with a 16 h light: 8 h dark photoperiod regimen. After ten days of culture, the rooted cuttings were transferred to plastic cones of 125 mm high and 32 mm diameter (M49, Polietilenos del Sur S.A. C.V.), one plant per cone. The substrate used was a 1:1 (v:v) mixture of vermiculite and sand. The substrate was autoclaved twice for 1 h at 121 °C and 15 psi. *S. rebaudiana* plants were inoculated with 150 spores of *R. irregularis* (M+) that were homogeneously distributed in the substrate; the controls were noncolonized (M-) plants. The plants were watered twice per week with 20 mL of half-strength Hoagland nutrient solution (*Hoagland & Arnon, 1950*) with $KH_2PO_4$ at the final phosphate concentrations that were evaluated: 20, 200, 500, and 1,000 μM. The pH of the nutrient solutions was adjusted to 6.1.

The plants were maintained in a growth chamber at 25 °C with a 16 h light:8 h dark photoperiod regimen for 30 days postinoculation (dpi). The experiment was performed

utilizing a complete factorial design, and six plants per phosphate concentration and colonization status with *R. irregularis* were evaluated. The controls were noncolonized plants treated with the different phosphate concentrations. Two independent experiments were performed. Similar trends were obtained in the both experiments, and the results of only one of them are shown.

## Staining and quantification of mycorrhizal colonization

*S. rebaudiana* root segments were stained with 0.05% trypan blue in lactoglycerol (*Phillips & Hayman, 1970*) and observed by light microscopy (BOECO Germany, BM-180) at 10-40X magnification. Total mycorrhizal colonization by *R. irregularis* was calculated according to the line-intersection method (*Giovannetti & Mosse, 1980*). For each plant, 90 root segments were assessed, and six plants were evaluated. The arbuscular percentage was calculated with MycoCalc software (https://www2.dijon.inrae.fr/mychintec/Mycocalc-prg/download.html). To identify the morphological type of the AM symbiosis in *S. rebaudiana*, mycorrhizal roots were stained with WGA-Alexa Fluor 488 to visualize the arbuscules, and the plant tissue was labeled with propidium iodide according to the methodology reported by *Xie et al. (2016)* using confocal laser scanning microscopy (LSM 800, Carl Zeiss).

## Determination of plant growth

The plants treated with the different phosphate concentrations and colonized (M+) or noncolonized (M-) with *R. irregularis* were collected at 30 days postinoculation (dpi). Shoots and roots of each plant were separated, total leaves number and the fresh weight of each organ was recorded. To minimize the dependency of the analysis of expression of genes and SG quantification with the leaf position on the plant. The leaves were collected in two equivalent groups, considering their opposite arrangement of the leaves and their position along the stem. Then, leaves positioned on each side of all the nodes of the plant composed each group.

The leaf area was determined by image analysis from leaves of the second node, close to the apical meristem. A stereomicroscope (Olympus SZX7, Germany) was used to obtain the micrographs. Image analysis was performed using ImageJ editing software (Version, 1.8.0.112).

One half of the leaves were frozen in liquid nitrogen for molecular analysis, and the other half for SG extraction.

Root were also collected and separated longitudinally in two sections, one of them was used for determination of mycorrhizal colonization, and the other for mycorrhiza-specific phosphate transporter identification. Six plants per phosphate treatment and *R. irregularis* colonization status were evaluated.

## Analysis of phosphorus and magnesium concentration

The leaves of *S. rebaudiana* plants were prepared according to *Guerrero-Molina et al. (2014)*. The leaves were analyzed by a high-resolution scanning electron microscope (SEM) equipped with a field emission cathode and coupled to an energy-dispersive X-ray (EDX, Carl Zeiss, Oberkochen, Germany). The electron energy used was 20 keV. The mapping of

P and Mg was determined by EDX to record the two-dimensional elemental composition of the leaf sample surface. For quantitative analyses, EDX spectrograms were recorded and analyzed using QUANTAX ESPRIT, Version 1.9 (BRUKER, Germany). Since the results of the concentration of each element is given as the percentage of such element with respect to all the components determined in the sample, the concentration of P and Mg in the samples was expressed as the percentage of each element in the leaves.

## Determination of chlorophyll and carotenoid concentration

The determination of chlorophyll and carotenoids concentration was made from three fresh leaves (approximately 100–150 mg) of each of the six plants of the M- and M + conditions. The leaves were collected from the upper, middle and lower part of each one of the analyzed plants and were ground in a mortar with 80% acetone. The extracts were centrifuged at 3,000 g for 15 min; the supernatants were separated, and the absorbance at 646.8, 663.2 and 470 nm was measured in a UV/Vis spectrophotometer (UV-1800, Shimadzu, Japan). The concentration of chlorophylls and carotenoids were calculated following the equations described by *Khan & Mitchell (1987)*.

## Measurement of chlorophyll fluorescence

The chlorophyll fluorescence was measured in the second fully expanded leaf of each plant using a chlorophyll fluorometer (model OS30P, Opti-Sciences Inc., USA). The evaluation was performed at room temperature according to the instructions for the chlorophyll fluorometer. Before the evaluation, the plants were placed in the dark for 30 min, and chlorophyll fluorescence was evaluated after applying a 1 s saturating pulse of actinic light ($3,500 \ \mu mol \ m^{-2} \ s^{-1}$). The primary fluorescence (Fo), maximal fluorescence (Fm), maximum quantum efficiency of PSII photochemistry (Fv/Fm), and potential photochemical efficiency (Fv/Fo) were calculated. Fv was calculated as Fv = Fm −Fo, and Fv/Fo was calculated as Fv/Fo = Fm/Fo −1 (*Schreiber, Bilger & Neubauer, 1994*).

## Expression analysis by qRT-PCR

The transcript accumulation levels of the genes for kaurene oxidase and (UDP)-glycosyltransferases were evaluated in colonized and noncolonized plants and in plants that were treated with 200 and 1,000 $\mu$M $KH_2PO_4$. These phosphate concentrations were selected because they were found to be the optimal and suboptimal conditions for inducing root colonization. Frozen leaves, from one of the groups described in the determination of plant growth section, were ground to a fine powder in liquid nitrogen. Total RNA was isolated from leaves using TRIzol reagent (Invitrogen, Carlsbad, CA) following the manufacturer's protocol. First-strand cDNA synthesis was performed as previously reported by *Cervantes-Gámez et al. (2016)*.

The primers used were those designed and reported by *Mandal et al. (2015)* for *S. rebaudiana* plants. The primers correspond to the kaurene oxidase gene (*SrKOF* 5′-TCTTCACAGTCTCGGTGGTG-3′, and *SrKOR* 5′-GGTGGTGTCGGTTTATCCTG-3′), the glucosyl transferase *UGT74G1* gene (*SrUGT74G1F* 5′- GGTAGCCTGGTGAAACATGG-3′, and *SrUGT74G1R* 5′- CTGGGAGCTTTCCCTCTTCT - 3′) and the glucosyl transferase *UGT76G1* gene (*SrUGT76G1F* 5′- GACGCGAACTGGAACTGTTG-3′, and *SrUGT76G1R*

5′- AGCCGTCGGAGGTTAAGACT - 3′). qRT-PCR was performed using SYBR$^{\circledR}$ Green (QIAGEN, USA) and quantified on a Rotor-Gene Q (QIAGEN, USA) real-time PCR thermal cycler. qRT-PCR was programmed for 35 cycles, with denaturing at 95 °C for 15 s, annealing at 55 °C for 30 s, and extension at 72 °C for 30 s. Primer specificity was verified by regular PCR and melting curve analysis. The primers for the *S. rebaudiana* glyceraldehyde-3-phosphate dehydrogenase (*GAPDH*) gene (*SrGAPDHF* 5′-TCAGGGTGGTGCCAAGAAGG-3′, and *SrGAPDHR* 5′- TTACCTTGGCAAGGGGAGCA - 3′) were used as internal controls for normalization, and the quantitative results were evaluated by the $2^{-\Delta\Delta CT}$ method described by *Livak & Schmittgen (2001)*. To interpret the results, genes with fold change values ≥1.5 were considered "upregulated", whereas genes with fold change values ≤ −0.7 were considered "downregulated". Six plants per phosphate treatment and *R. irregularis* colonization status were evaluated.

## Cloning the mycorrhiza-specific phosphate transporter gene from *S. rebaudiana*

A pair of degenerate primers (*SrPTF* 5′- ATGGGDTTTTTYACYGATGC-3′and *SrPTR* 5′- GGNCCAAARTTSGCRAAGAA- 3′) were designed by aligning highly conserved regions of AM-specific phosphate transporters from *M. truncatula* (accession number: AY116210), *A. sinicus* (accession number: JQ956418), *S. lycopersicum* (accession number: AF022874), *S. tuberosum* (accession number: AY793559) and *P. hybrida* (accession number: EU532763). The PCR product was purified using the QIAquick PCR Purification Kit (QIAGEN, USA) and ligated into the pGEM$^{\circledR}$-T Easy vector (Promega, USA) in accordance with the manufacturer's protocols. The presence of the correct insert (1350 bp) within the pGEM$^{\circledR}$-T Easy vector was confirmed by PCR using the universal primers T7 and SP6, and the insert was then sequenced.

Collected roots from one of the groups described in the determination of plant growth section, were ground to a fine powder in liquid nitrogen. Total RNA was obtained from the roots of six colonized (M+) and six noncolonized (M-) plants fertilized with 200 μM KH$_2$PO$_4$. RNA was isolated using TRIzol reagent (Invitrogen, Carlsbad, CA) following the manufacturer's protocol. First-strand cDNA synthesis was performed as previously reported by *Cervantes-Gámez et al. (2016)*. cDNA synthesis was confirmed by PCR using primers for the *S. rebaudiana* glyceraldehyde-3-phosphate dehydrogenase (*GAPDH*) gene (*SrGAPDHF* 5′- ATGGGDTTTTTYACYGATGC-3′and *SrGAPDHR* 5′-GGNCCAAARTTSGCRAAGAA- 3′).

For the expression analysis of *SrPT* in the M- and M+ plants, PCRs were run in a total reaction volume of 10 μL, comprising 0.2 μL of GoTaq$^{\circledR}$ Flexi DNA Polymerase (Promega, USA), 200 nM of each primer, and 50 ng of cDNA. The PCR thermocycler was programmed for 35 cycles, with denaturing at 95 °C for 15 s, annealing at 52 °C for 1 min, and extension at 72 ° C for 30 s. The *SrGAPDH* gene was used as the reference gene.

## Homology modeling analysis of the AM-specific phosphate transporter from *S. rebaudiana*

BLAST analysis of the SrPT gene sequence was performed to determine homology predictions using the tools on the NCBI website (https://www.ncbi.nlm.nih.gov). To

determine the conserved region of SrPT, multiple sequence alignments of AM-specific phosphate transporter proteins were performed using MULTALIN software. The homology model of SrPT transmembrane domains (TDs) was constructed according to *Yadav et al. (2010)*, and the Mtpt4 protein structure from *M. truncatula* was used as the template for homology modeling for the *S. rebaudiana* AM-specific phosphate transporters.

### Steviol glycosides extraction and quantification of concentration

The leaves of colonized (M+) and noncolonized (M-) plants that were treated with 200 and 1,000 $\mu$M $KH_2PO_4$ were used to evaluate the SG concentration. Leaves from one of the groups described in the determination of plant growth section, were dried in in an oven (Thermo Scientific, USA) at 65 °C for 48 h. The dry leaf tissue (0.1 g) was extracted with one mL of methanol (J.T. Backer, USA), following the methodology described by *Woelwer-Rieck et al. (2010)*. The mixture was stirred for 3 min, allowed to stand for 24 h without stirring, and then centrifuged at 10,000 rpm at 4 °C for 10 min. The supernatant was recovered, placed in Eppendorf tubes, and stored at −4 °C until the analysis by HPTLC (CAMAG, Switzerland). The quantification of SGs was based on the methodology reported by *Bladt & Zgainski (1996)* and *Morlock et al. (2014)*, and described recently by *Villamarin-Gallegos et al. (2020)*. Stevioside and rebaudioside A concentration were expressed in mg g $DW^{-1}$. Six plants per phosphate treatment and *R. irregularis* colonization status were evaluated.

### Statistical analysis

The differences between the total colonization and arbuscular percentages were examined by one-way analysis of variance (ANOVA), and Tukey's post hoc test ($P < 0.05$) was performed to test the significance of differences between means. Data regarding the effect of the interaction between the $KH_2PO_4$ concentration and mycorrhizal colonization on plant growth, P and Mg concentration, pigment concentrations, chlorophyll fluorescence and SG concentration were subjected to factorial two-way analysis of variance (ANOVA). Tukey's post hoc test was used to analyze the differences. The paired Student's $t$-test was used to evaluate the significance of differences in the gene expression of kaurene oxidase and (UDP)-glycosyltransferases. All data used for ANOVA and factorial analysis were checked for normal distributions (Shapiro–Wilk's test) before statistical analysis. All statistical analyses were performed using the statistical software IBM SPSS for Windows, Version 24.0 (Armonk, NY, IBM Corp.).

## RESULTS

### Phosphate concentration affects the mycorrhizal colonization of *S. rebaudiana*

The highest percentages of colonization were obtained at 20 and 200 $\mu$M $KH_2PO_4$ (73.3 and 67.0% colonization, respectively). In contrast, the percentage of total colonization decreased significantly at 500 and 1,000 $\mu$M $KH_2PO_4$, with 43.3 and 18.4% colonization, respectively (Fig. 1). The percentage of arbuscules was significantly reduced at 500 and 1,000 $\mu$M $KH_2PO_4$, with 1.48 and 0.4%, respectively (Fig. 1).

In the roots of plants treated with 20 $\mu$M $KH_2PO_4$, a high number of arbuscules formed (Fig. 2A, see label *); several intraradical hyphae grew through the cortical cells (Fig. 2A,

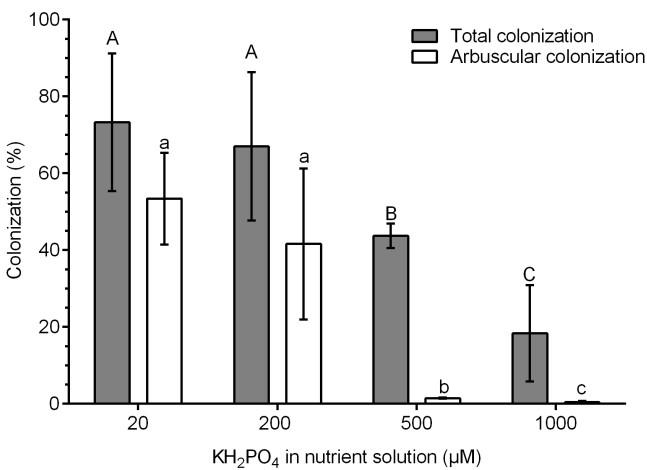

**Figure 1** **Quantification of mycorrhizal colonization in *S. rebaudiana* roots.** Percentages of total mycorrhizal colonization and arbuscule colonization by *R. irregularis* in *S. rebaudiana* plants fertilized with Hoagland nutrient solution at different $KH_2PO_4$ concentrations. Bars represent the mean $\pm$ standard deviation (SD) of six replicates. Different letters indicate significant differences according to Tukey's test ($P < 0.05$). Capital letters were used for total colonization, and lowercase letters were used for arbuscular colonization.

see label **ih**), although vesicle structures were scarce. Similar structures were observed in colonized plants (M+) with 200 µM $KH_2PO_4$; however, under these experimental conditions, the formation of arbuscules and vesicle structures was more evident (Fig. 2B, see labels ⋆ and **v**), indicating that 200 µM $KH_2PO_4$ created better conditions than 20 µM for the formation of arbuscules. Qualitative differences in mycorrhizal structures were observed when plants were treated with the highest $KH_2PO_4$ concentrations (500 and 1,000 µM); the formation of arbuscules, for instance, was significantly reduced (Figs. 2C and 2D, see label ⋆), and shortening of intraradical structures was observed (Figs. 2C and 2D, see label **ih**) in comparison to the structures observed at low $KH_2PO_4$ concentrations (Figs. 2A and 2B).

The activation of specific genes, such as the phosphate transporter specifically induced by the mycorrhizal association, is an important marker for evaluation successful colonization establishment. To our knowledge, there is no information on this transporter type in *S. rebaudiana*. For this reason, we identified a putative mycorrhiza-specific phosphate transporter in *S. rebaudiana* (*SrPT*) by a simple PCR strategy based on degenerate oligonucleotides to clone the corresponding phosphate transporter. A 1175 bp-long genomic fragment containing an open reading frame that encodes a 391-amino acid polypeptide with a molecular mass of 43.39 kDa was cloned (accession number: MN273502). This putative SrPT polypeptide contains 9 of the 12 transmembrane domains from the canonic phosphate transporter (Fig. S1). The bioinformatic analysis suggests that SrPT is 72.89% conserved in comparison to the sequences reported for MtPT4 TMDs in *Medicago truncatula*. Notably, *SrPT* transcript accumulation increased in the roots of colonized (M+) plants compared with that in the roots of noncolonized (M-) plants. This

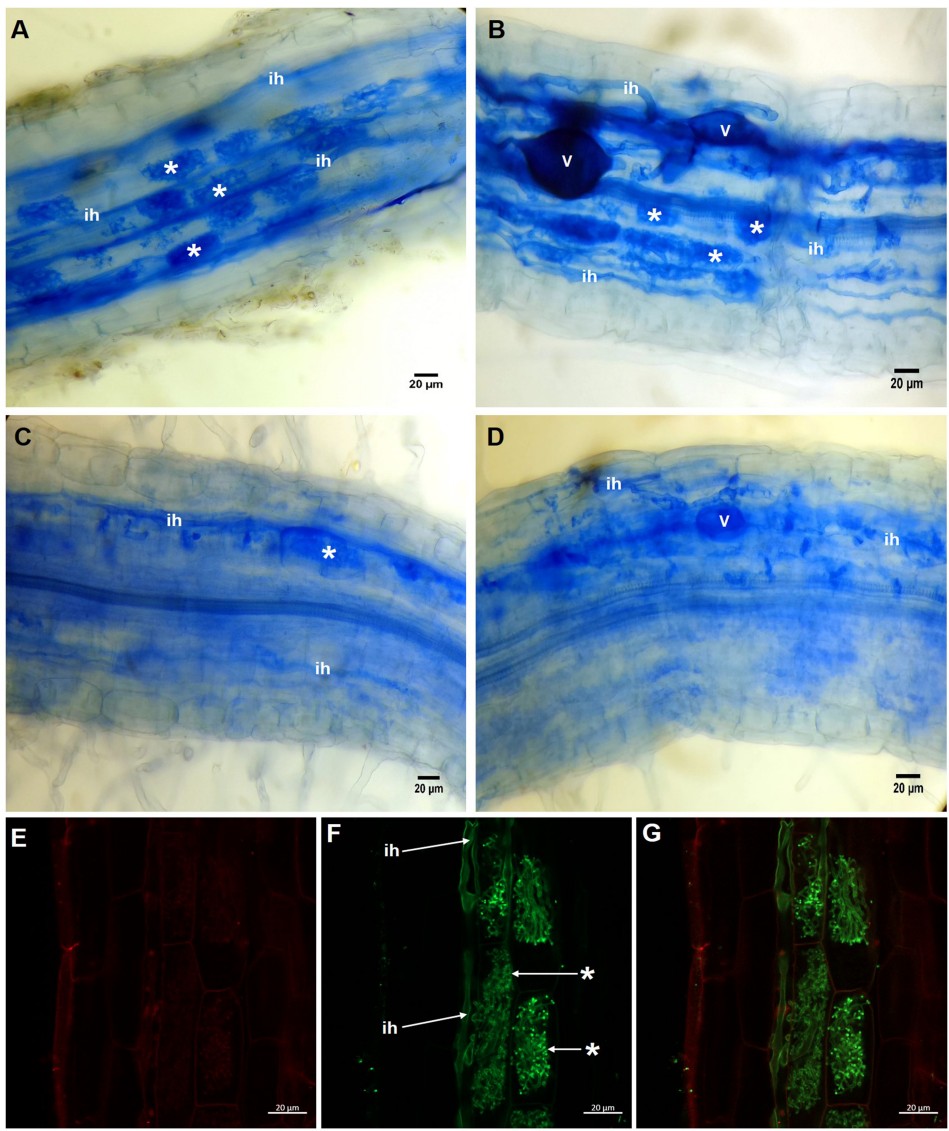

**Figure 2** **Light and confocal microscopic analysis of *R. irregularis* colonization structures in *S. rebaudiana* roots.** Mycorrhiza-colonized roots of *S. rebaudiana* plants fertilized with 20 (A), 200 (B), 500 (C) and 1,000 $\mu$M KH$_2$PO$_4$ (D) after trypan blue staining depicting the arbuscules, vesicles and intraradical hyphae. Mycorrhiza-colonized roots with 200 $\mu$M KH$_2$PO$_4$ were also treated with propidium iodide to label the cell wall (E) and with WGA-Alexa Fluor 488 conjugate to stain the fine details of the intraradical hyphae and arbuscules (F and G). The merged image showing both red and green fluorescence is presented in (G). Vesicles: V; intraradical hyphae: ih; arbuscules: *.

result suggests that this gene is positively regulated by AM symbiosis in *S. rebaudiana*, in a similar manner to the genes for other mycorrhiza-specific phosphate transporters reported in other model plants (Fig. S2).

Confocal microscopic analysis of the colonized roots and staining with the conjugate WGA-Alexa Fluor® 488 and propidium iodide as fluorescent markers permitted us to differentiate the hyphae and the plant cells, respectively. Intracellular hyphae and the

formation of arbuscules from intracellular hyphae growing in the inner cortex were observed (Figs. 2E–2G, see labels **ih** and **\***). With this approach, we were able to depict and classify this structure as a typical *Arum*-type mycorrhiza.

## Mycorrhizal colonization improves plant growth in *S. rebaudiana*

In the M- plants, the leaf fresh weight was not different than that in plants treated with 20, 200 and 500 $\mu$M $KH_2PO_4$; the leaf fresh weight only increased significantly at 1,000 $\mu$M $KH_2PO_4$ (Fig. 3A). In the M+ plants, the leaf fresh weight increased by a factor of 1.74 with 200 $\mu$M $KH_2PO_4$ in comparison to that in the M- plants (control) at the same phosphate concentration. However, no difference was found between M+ and M- plants at 1,000 $\mu$M $KH_2PO_4$. The leaves of M+ plants with 200 $\mu$M $KH_2PO_4$ had a similar fresh weight to those of M+ and M- plants treated with 1,000 $\mu$M $KH_2PO_4$ (Fig. 3A). In plants treated with 200 $\mu$M $KH_2PO_4$, the fresh weight of roots was higher in the roots of M+ plants than in the roots of M-plants. At 1,000 $\mu$M $KH_2PO_4$, the fresh weight of roots was higher in the M+ plants than in the M- plants (Fig. 3B). Leaf number and foliar area were determined, and only at 200 $\mu$M $KH_2PO_4$, M- plant showed fewer leaves than M+, as well as foliar area (Fig. S3), which is consistent with the pattern of fresh weight of leaves. Foliar area, on the other hand, was only significantly lower in M+ plants than M- plants at 500 $\mu$M $KH_2PO_4$ (Fig. S3).

## The phosphorus and magnesium concentration increases in the leaves of colonized plants with a low phosphate concentration

The P concentration was four times higher in the leaves of M+ plants treated with 200 $\mu$M $KH_2PO_4$ than that in M- plants. At 500 and 1,000 $\mu$M $KH_2PO_4$, the P concentration was similar in the leaves of M+ plants and M- plants (Fig. 4A). In the leaves of M- plants, the Mg concentration increased significantly at 500 and 1,000 $\mu$M $KH_2PO_4$, while in the leaves of M+ plants, the Mg concentration increased at 20, 200 and 1,000 $\mu$M $KH_2PO_4$, but the Mg concentration was lower at 500 $\mu$M $KH_2PO_4$ (Fig. 4B). These results suggest that AM symbiosis can stimulate P and Mg accumulation at low $KH_2PO_4$ concentrations.

## Chlorophyll fluorescence and the concentration of photosynthetic pigments improve in colonized plants at a low phosphate concentration

Chlorophyll fluorescence was used as an indicator of photosynthetic performance in the *S. rebaudiana* plants. In M- plants and at all $KH_2PO_4$ concentrations, the Fv/Fm ratio values were less than 0.8. In M+ plants at 20 and 200 $\mu$M $KH_2PO_4$, the Fv/Fm ratio values were greater than 0.8, and at 500 and 1,000 $\mu$M $KH_2PO_4$, the Fv/Fm ratio values diminished to less than 0.8 (Fig. 5A). The Fv/Fo ratio is indicative of the photochemical efficiency of photosynthesis. In the M- plants at all phosphate concentrations, the values of the Fv/Fo ratio were less than 4.0. In M+ plants at 20 and 200 $\mu$M $KH_2PO_4$, the value of the Fv/Fo ratio was greater than 4.0; at 500 and 1,000 $\mu$M $KH_2PO_4$, this ratio was less than 4.0 (Fig. 5B). The values of Fo, Fm and Fv are presented in Fig. S4.

The total concentration of chlorophylls and carotenoids did not change in the M- plants at any phosphate concentration. However, in M+ plants at 20 and 200 $\mu$M $KH_2PO_4$, the

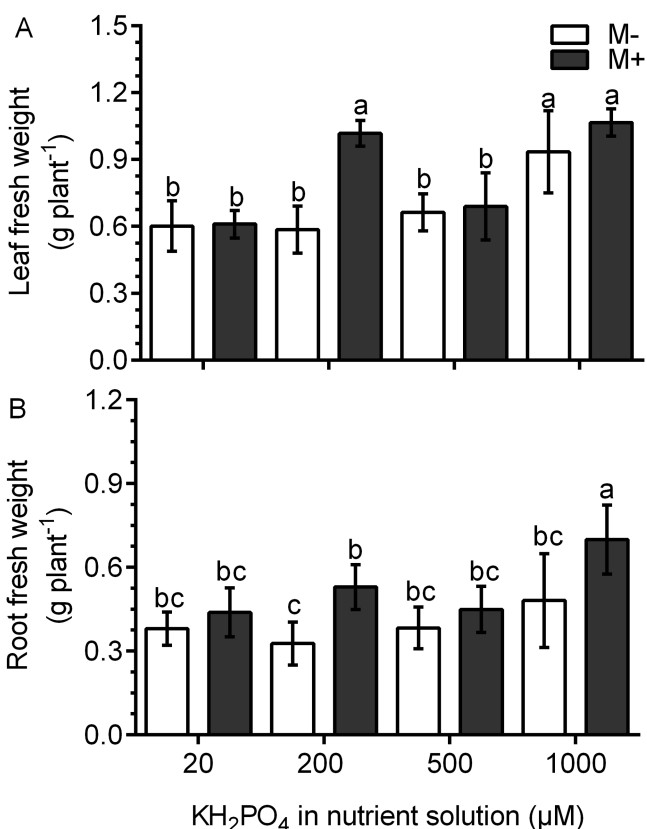

**Figure 3** **Effect of AM symbiosis and $KH_2PO_4$ concentrations on the fresh weight of *S. rebaudiana* roots and leaves.** Fresh weight of leaves (A) and roots (B) of mycorrhiza-colonized (M+) and noncolonized (M-) *S. rebaudiana* plants fertilized with Hoagland solution with different $KH_2PO_4$ concentrations. Bars represent the mean ± standard deviation (SD) of six replicates. Different letters indicate significant differences according to Tukey's test ($P < 0.05$).

concentration of chlorophylls and carotenoids was higher as compared to the M- plants at the same phosphate concentrations; at 500 and 1,000 µM $KH_2PO_4$, the concentration of the two pigments were similar in M- and M+ plants (Figs. 5C and 5D).

## Differential expression of the genes for kaurene oxidase and glucosyltransferases in colonized plants with added phosphate

The transcription of the kaurene oxidase (*KO*) gene was increased 7.5 times in M+ plants at 200 µM $KH_2PO_4$ compared with that in M+ plants without added $KH_2PO_4$. The transcription level of the *KO* gene did not change in M+ plants at 1,000 µM $KH_2PO_4$ in comparison to M- plants, since the relative expression ($2^{-\Delta\Delta Ct}$) was close to 1 (Fig. 6A). The *UGT74G1* gene encoding the protein involved in stevioside synthesis was upregulated in M+ plants at 200 µM $KH_2PO_4$; the level of relative expression was over 1.5 (Fig. 6B). The *UGT76G1* gene encoding the protein involved in rebaudioside A synthesis was downregulated in M+ plants at the same $KH_2PO_4$ concentration, and its relative expression was less than 0.7 (Fig. 6C). However, in M+ plants at 1,000 µM $KH_2PO_4$, the expression

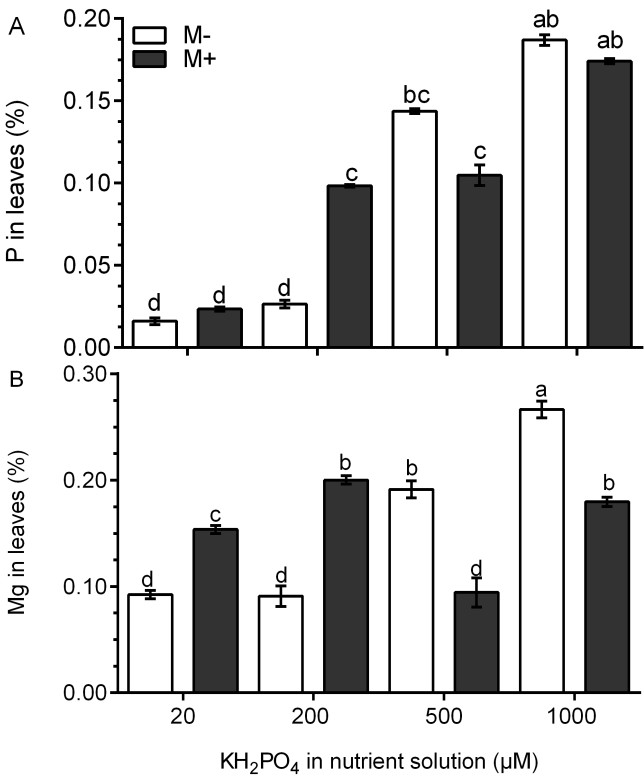

**Figure 4  Quantification of P and Mg under AM symbiosis and different $KH_2PO_4$ concentrations.** Phosphorus (A) and magnesium (B) concentration in the leaves of mycorrhiza-colonized (M+) and noncolonized (M-) *S. rebaudiana* plants fertilized with Hoagland solution at different $KH_2PO_4$ concentrations. Bars represent the mean ± standard deviation (SD) of six replicates. Different letters indicate significant differences according to Tukey's test ($P < 0.05$).

of the *UGT74G1* gene was downregulated, and the expression of the *UGT76G1* gene was unchanged (Figs. 6B and 6C).

## SGs differentially accumulate in the leaves of colonized plants with added phosphate

In M+ plants at 200 μM $KH_2PO_4$, the stevioside concentration was 2.8 times higher and the rebaudioside A concentration was 1.61 times lower than those of M- plants (Figs. 7A and 7B). This metabolite accumulation is consistent with the transcript levels of the corresponding glucosyl transferases (Figs. 6B and 6C). At 1,000 μM $KH_2PO_4$, the accumulation of the two metabolites in M+ and M- plants was not affected (Figs. 7A and 7B).

## DISCUSSION

AM symbiosis enhances P uptake in many plants and plays an important role in agricultural and natural environments (*Smith et al., 2011*). Phosphate availability may change the initial signaling for the establishment, maintenance, and functioning of AM symbiosis (*Schmitz & Harrison, 2014*). In this study, the low phosphate concentrations (20 and 200 μM

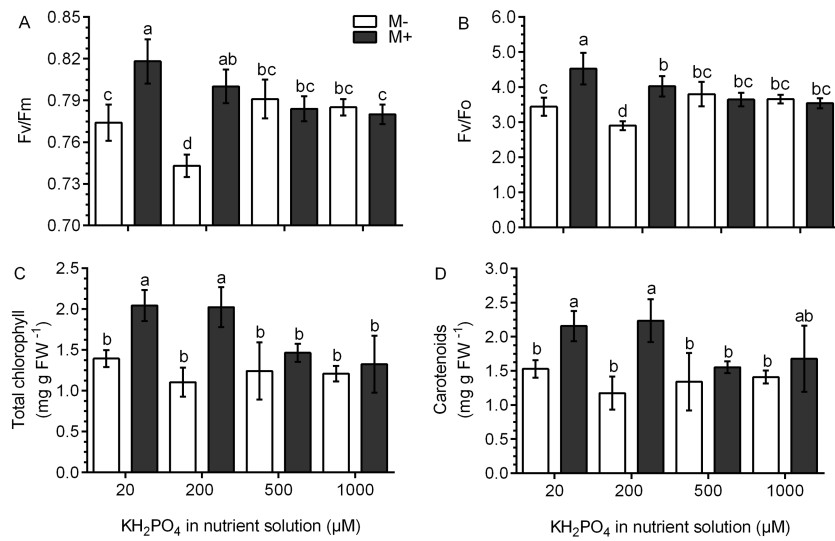

**Figure 5** **Effects of AM symbiosis and $KH_2PO_4$ concentrations on photosynthetic performance in *S. rebaudiana*.** Maximal photochemical efficiency (Fv/Fm) (A), potential photochemical efficiency (Fv/Fo) (B), and total chlorophyll (C), and carotenoid contents (D) in leaves of mycorrhiza-colonized (M+) and noncolonized (M-) *S. rebaudiana* plants fertilized with Hoagland solution at different $KH_2PO_4$ concentrations. Fv/Fm and Fv/Fo ratios were obtained from chlorophyll fluorescence measurements. Bars represent the mean ± standard deviation (SD) of six replicates. Different letters indicate significant differences according to Tukey's test ($P < 0.05$).

$KH_2PO_4$) stimulated a high percentage of total mycorrhizal colonization by *R. irregularis* in *S. rebaudiana* plants, while the high $KH_2PO_4$ concentrations (500 and 1,000 µM) decreased the colonization efficiency of *R. irregularis* in *S. rebaudiana* plants by approximately 30 and 70%, respectively. AM symbiosis is inhibited by a high concentration of $KH_2PO_4$ (*Smith, Smith & Jakobsen, 2004*), and the sensitivity and inhibition percentage depend on the plant species and AM fungus. In *Medicago truncatula*, fertilization with 1.3 mM phosphate reduced AM symbiosis by 80% compared to that in plants fertilized with 0.13 mM phosphate (*Bonneau et al., 2013*). In *Petunia hybrida*, phosphate at 100 µM induces high AM symbiosis, while phosphate at 3 mM and higher concentrations completely suppressed this symbiosis (*Nouri et al., 2014*). Therefore, it was important in our study to define this effect of phosphate on the colonization of *R. irregularis* in *S. rebaudiana* plants.

The formation of arbuscules in *S. rebaudiana* roots was inhibited to a higher extent than the total colonization, indicating that arbuscule formation is more sensitive to high phosphate concentrations than other fungal structures, such as hyphae and vesicles. In addition, shortening of intraradical structures was observed in comparison to the arbuscules of roots at low phosphate concentrations. Similarly, changes in arbuscule structures were observed in *P. hybrida* plants; fertilization with high phosphate concentrations significantly reduced the development of arbuscules and resulted in malformed arbuscules with fewer branches (*Breuillin et al., 2010*).

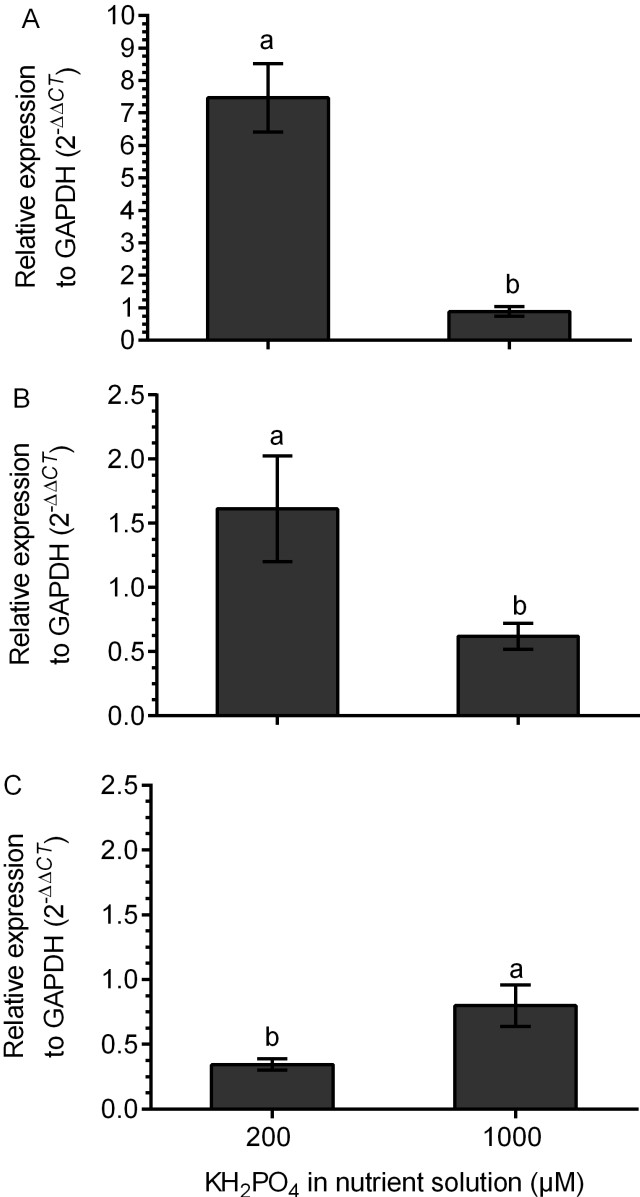

**Figure 6** **Differential transcript accumulation of three key genes from the SG biosynthetic pathway under AM symbiosis and different KH₂PO₄ concentrations.** *KO* (A), *UGT74G1* (B) and *UGT76G1* (C) relative expression levels in *S. rebaudiana* plants that were mycorrhized and fertilized with Hoagland solution at 200 and 1,000 µM KH₂PO₄. For each condition, the transcript levels of the *KO*, *UGT74G1* and *UGT76G1* genes were first normalized against *SrGAPDH* and then normalized against the gene expression of *S. rebaudiana* without inoculation. The analysis of the relative gene expression data used the $2^{-\Delta\Delta CT}$ method. Bars represent the mean ± standard deviation (SD) of three biological and three technical replicates. Different letters indicate significant differences according to Student's *t* test ($P < 0.05$).

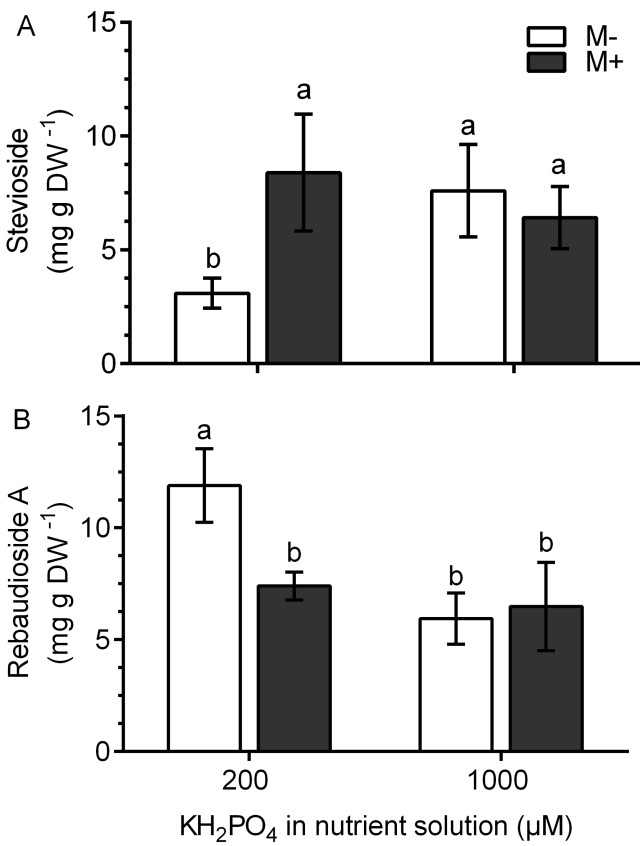

**Figure 7** **Quantification of SG concentration under AM symbiosis and different KH$_2$PO$_4$ concentrations.** Stevioside (A) and rebaudioside A (B) concentration in mycorrhiza-colonized (M+) and noncolonized (M-) *S. rebaudiana* plants fertilized with Hoagland solution at 200 and 1,000 μM KH$_2$PO$_4$ . Bars represent the mean ± standard deviation (SD) of six replicates. Different letters indicate significant differences according to Tukey's test ($P < 0.05$).

Previous studies in *S. rebaudiana* have reported colonization by *R. irregularis*, but the AM colonization morphology type has not been documented (*Vafadar, Amooaghaie & Otroshy, 2014*; *Mandal et al., 2015*; *Tavarini et al., 2018*). In this study, confocal microscopic analysis with specific fluorescent dyes indicated that the AM colonization is classified as an *Arum*-type morphology. This AM morphotype is highly sensitive to environmental factors, including soil nutrients (*Dickson et al., 2003*). It is also suggested that *Arum*-type colonization is more efficient than other morphotypes in the acquisition and transference of phosphate from the soil to the plant, resulting in better plant growth (*Van Aarle et al., 2005*)

The beneficial effects of AM symbiosis on growth promotion and yield have been reported in plants from the Asteraceae family (*Rapparini, Llusià & Peñuelas, 2008*; *Aroca et al., 2013*). In this study, AM symbiosis with 200 μM KH$_2$PO$_4$ increased colonization and arbuscule formation but also promoted the leaf and root growth in *S. rebaudiana* plants; the growth of these plants was comparable to the growth of plants fertilized at higher KH$_2$PO$_4$ concentrations (1,000 μM). These results suggest the activation of a high-affinity

mycorrhiza-specific phosphate transporter during AM symbiosis, as reported for other plant species. In fact, we found the accumulation of transcripts of a putative phosphate transport gene in response to mycorrhizal colonization. This is the first time that a putative phosphate transporter gene has been identified in *S. rebaudiana* plants, and we anticipate that this gene encoding the specific phosphate transporter may contribute to growth promotion in leaves and roots. These results indicate that colonized plants are able to compensate for the deficiency in P nutrition using the mycorrhizal P uptake pathway. Similar effects have been reported in *M. truncatula* and *P. hybrida* plants at different phosphate concentrations (*Balzergue et al., 2013*; *Nouri et al., 2014*). Shoots and roots were responsive to mycorrhizal colonization by increasing growth at 200 $\mu$M $KH_2PO_4$, whereas at 1,000 $\mu$M $KH_2PO_4$, roots growth was stimulated only in M+ compared to noncolonized controls (M-). This indicates that the symbiosis differentially affects shoot and root growth, and this depends on the phosphate fertilization regime. In plants at 500 $\mu$M $KH_2PO_4$, such differences between fresh weights of shoots and roots in M+ in comparison to M- was not observed. This in contrast to 200 $\mu$M and 1,000 $\mu$M $KH_2PO_4$ conditions, in which mycorrhiza-specific and direct plant phosphate uptake pathways dominate the phosphate uptake, respectively. It is possible that 500 $\mu$M $KH_2PO_4$, the direct and the mycorrhiza-specific phosphate uptake pathways, may be interacting in such a way that no increased in growth is manifested. Although this hypothesis needs to be further studied.

Chlorophyll fluorescence is used as an indicator of light assimilation (electron transport) in the reaction centers of chlorophyll and as an indirect measurement of photosynthetic performance in plants subjected to different conditions of stress (*Schreiber, Bilger & Neubauer, 1994*; *Harbinson, 2013*). Stress conditions may disrupt components of the photosynthetic apparatus and affect photosynthetic performance. In healthy plants, the values of the Fv/Fm ratio are between 0.79 and 0.82; however, under stress conditions, the photosynthetic performance is affected, and these values decrease to below 0.79. Likewise, values of the Fv/Fo ratio between 4-5 correspond to normal values for healthy plants, and values less than 4 are indicative of a loss of photosynthetic efficiency that may result from a stress condition (*Baker, 2008*). In the noncolonized *S. rebaudiana* plants at low $KH_2PO_4$ concentrations, these values were less than 0.79 and 4.0, which indicates that the plants are subjected to stress conditions. However, these values in colonized plants at the same $KH_2PO_4$ concentration were significantly higher than those in noncolonized plants. These results suggest that AM symbiosis may compensate for the nutritional stress induced by low $KH_2PO_4$ concentrations by restoring the electron transfer through PSII, regulating the functionality of the reaction center, and improving photosynthetic performance in *S. rebaudiana* plants. The results are consistent with those of other studies performed in plants under stress conditions such as high salinity, high temperatures, and water stress (*Sheng et al., 2008*; *Zhu et al., 2011*; *Hu et al., 2017*).

In colonized *S. rebaudiana* plants at low phosphate concentrations, the concentration of photosynthetic pigments was higher than that in noncolonized plants at the same phosphate concentration. Similarly, other authors have shown that mycorrhizal colonization

stimulates the accumulation of these compounds in plants (*Colla et al., 2008*; *Nafady & Elgharably, 2018*). These studies support the idea that photosynthesis is improved in colonized plants, which ensures carbon fixation and provides a carbon source to the fungal symbiont under low phosphate conditions (*Zai et al., 2012*). Additionally, increased carotenoid concentration in plants is associated with light harvesting, photoprotection, and antioxidant processes, which may contribute to improving plant growth (*Walter, 2013*).

Mg is bound to the central atom in the porphyrin ring of chlorophyll a and b, and 25–60% of the total Mg in plants exists as chlorophyll-bound Mg (*Chen et al., 2018*). A significant increase in the Mg percentage was found in the leaves of *S. rebaudiana* plants colonized under the low phosphate concentration. This result is consistent with the increase in chlorophyll concentration and has been reported in other plants (*Colla et al., 2008*; *Vafadar, Amooaghaie & Otroshy, 2014*). Thus, the increase in the concentration of chlorophylls, Mg, and carotenoids improves plant growth and metabolite accumulation, confirming the positive effect of AM symbiosis.

Kaurene oxidase (KO) plays an important role in SG biosynthesis and represents an important branch point that specifically directs the flow of metabolites towards the biosynthesis of steviol (the central backbone of SGs). In fact, SGs are synthesized from steviol glycosylation, where the conjugation of glucose to steviol is carried out by UDP-glycosyltransferases (UGTs); for example, UGT74G1 is known to convert steviolbioside to stevioside, and UGT76G1 adds the final glucose required to produce rebaudioside A (*Brandle & Telmer, 2007*).

The *KO* and *UGT74G1* gene transcription levels were upregulated in colonized plants compared to those in noncolonized plants at 200 $\mu$M $KH_2PO_4$, which was consistent with the observed stevioside accumulation trends. In contrast, in noncolonized plants at 200 $\mu$M $KH_2PO_4$, the expression of the *UGT76G1* gene and the corresponding accumulation of rebaudioside A were higher than those in colonized plants. These results indicate that plant colonization with AM stimulates stevioside accumulation, while in noncolonized plants, the accumulation of rebaudioside A is stimulated. In addition, the high phosphate concentration (1,000 uM $KH_2PO_4$) in colonized plants downregulated the expression of the *UGT74G1* gene and did not change the expression of the *KO* or *UGT76G1* genes at the same phosphate concentration. These results support the idea that SG synthesis is sensitive to the phosphate concentration and mycorrhizal interactions and suggest that AM symbiosis causes an increase in stevioside accumulation by modulating the expression of the *KO* and *UGT74G1* genes, which convert steviolbioside to stevioside. The downregulation of the *UGT76G1* gene, which encodes a protein involved in transforming stevioside to rebaudioside A, may also contribute to the accumulation of stevioside. In contrast, the accumulation of rebaudioside A, which is the SG with the highest sweetening power, would be favored in noncolonized plants. This information is relevant from a biotechnological perspective. The results demonstrate the effect of the phosphate concentration on the mycorrhizal interaction between *S. rebaudiana* and *R. irregularis* as well as the effect on SG concentration after the modulation of the expression of key biosynthetic genes.

## CONCLUSIONS

AM symbiosis between *S. rebaudiana* and *R. irregularis* is affected by phosphate concentrations; a low phosphate concentration induces a high percentage of colonization. The morphology of the colonization structure was a typical *Arum*-type mycorrhiza, and a mycorrhiza-specific phosphate transporter was identified. Colonization at low phosphate concentrations improved plant growth, the chlorophyll and carotenoid concentrations, and photochemical performance. The low phosphate concentration improved mycorrhizal colonization and modulated the stevioside and rebaudioside A concentration by regulating the transcription of genes that encode kaurene oxidase and glucosyltransferases, which are involved in the synthesis of these compounds in *S. rebaudiana*. This knowledge is important for generating biotechnological strategies that involve manipulating the concentration of stevioside or rebaudioside A by controlling the colonization status and the phosphate concentration of *S. rebaudiana* plants.

## ACKNOWLEDGEMENTS

We owe special thanks to Silvia Evangelista Lozano from CeProBi-IPN for providing *Stevia rebaudiana* plants. The authors thank Daniel Tapía Maruri for the excellent technical assistance in producing confocal laser scanning microscope images.

### Funding

The authors received funding from the Consejo Nacional de Ciencia y Tecnología (Curriculum Vitae Unico: 480787) and Secretaría de Investigación y Posgrado-Instituto Politécnico Nacional (Beca de Estímulo Institucional de Formación de Investigadores). This work was conducted with the support of Secretaría de Investigación y Posgrado-Instituto Politécnico Nacional (projects: 20180427, 20181785, 20195064 and 20200699) and Dirección General de Asuntos del Personal Académico IN209118 and CV200519. The funders had no role in study design, data collection and analysis, decision to publish, or preparation of the manuscript.

### Grant Disclosures

The following grant information was disclosed by the authors:
Consejo Nacional de Ciencia y Tecnología: Curriculum Vitae Unico: 480787.
Secretaría de Investigación y posgrado-Instituto Politécnico Nacional (Beca de Estímulo Institucional de Formación de Investigadores).
Secretaría de Investigación y posgrado-Instituto Politécnico Nacional: 20180427, 20181785, 20195064, 20200699.
Dirección General de Asuntos del Personal Académico: IN209118, CV200519.

### Competing Interests

The authors declare there are no competing interests.

## Author Contributions

- Luis G. Sarmiento-López conceived and designed the experiments, performed the experiments, prepared figures and/or tables, authored or reviewed drafts of the paper, and approved the final draft.
- Melina López-Meyer analyzed the data, prepared figures and/or tables, authored or reviewed drafts of the paper, and approved the final draft.
- Gabriela Sepúlveda-Jiménez analyzed the data, authored or reviewed drafts of the paper, and approved the final draft.
- Luis Cárdenas and Mario Rodríguez-Monroy conceived and designed the experiments, authored or reviewed drafts of the paper, and approved the final draft.

## DNA Deposition

The following information was supplied regarding the deposition of DNA sequences:

Mycorrhiza-specific phosphate transporter from *S. rebaudiana* sequence are available at GenBank: MN273502.

## Data Availability

The raw data are available in the Supplemental Files.

## Supplemental Information

Supplemental information for this article can be found online at http://dx.doi.org/10.7717/peerj.10173#supplemental-information.

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
