# Peer review of "Photosynthetic performance and stevioside concentration are improved by the arbuscular mycorrhizal symbiosis in Stevia rebaudiana under different phosphate concentrations"

_PeerJ, doi:10.7717/peerj.10173_

## Round 0.1 · original submission · Major Revisions

Dear Dr. Sarmiento López and Dr. Rodríguez Monroy,

Please find below the comments of two independent reviewers. Although the MS addresses a scientifically sound question, and is well structured, in my opinion, it requires major changes. Please pay particular attention to the experimental design, missing data and interpretation of the results.

Looking forward to receiving the revised MS.

Sincerely,
Ana I. Ribeiro-Barros

Reviewer 1 ·

Basic reporting

See below

Experimental design

See below

Validity of the findings

See below

Additional comments

Report on ‘Phosphate modulates arbuscular mycorrhizal symbiosis and improves photosynthetic performance and stevioside content in Stevia rebaudiana’

This is a very well written, clear manuscript, that builds on those of the cited publications of Mandal et al. Vafadar et al., Tavarini et al. It deserves publishing because it provides evidence for the type of colonisation structure, for the mode of uptake of PO4 and usefully links the data on photosynthesis [fluorescence and pigments] with growth and SGs [but see my comment below ref Reb-A and stevioside concentrations.
A number of small issues need to be addressed, and then I believe it will be suitable for publication.
I have annotated the text [attached], and herein indicate some of the more important issues that must be addressed.
Title: as it read it implies that phosphate improves photosynthetic performance and stevioside… so either add ‘the latter’ or rewrite completely.
Also, I disagree with the use throughout of ‘content’ when indeed the authors mean concentration’ For content = concentration * mass, and at no time is mass information given in the text. So, change all uses of ‘content’ to ‘concentration’ [except of course in the references!].
Line 52 Rhizoglomus irregulare or Rhizophagus irregularis I am not an expert on this but ensure the correct one is used.
See line 97, do not start a sentence with an acronym [and define at first use in the text [separate from the Abstract].
Likewise, Line 117.
A little more detail in line 138.
Line 345 any reason the root fwt was higher in M+ than M- at 1000 µM?
Line 383 compared to what..? Be explicit.
Line 460 indicate if significant or not.
Lines 497-509 need some clarification. For a start, Reb-A is more preferable from a taste perspective than stevioside, and indeed the total SGs is the same with or without M+ at 200 µM. So this needs some extra discussion. And reconcile lines 499 and 505/6.

Annotated reviews are not available for download in order to protect the identity of reviewers who chose to remain anonymous.

Reviewer 2 ·

Basic reporting

The English version is acceptable, although some modifications are required.

The literature references are not always referring to the original literature, e.g., for the Synthesis of steviol which has been published by Totté et al. already in 2000

The structure of the paper is OK

The results are sometimes rather weak, due to the lack of repetitions of the whole experiment.

Experimental design

The intention of the authos is good, but the way that they organised the experiment is not. Only 6 plants per treatment were used, but interesting information on the plants is missing, e.g., the number of leaves, the position of the leaves that were used. It is not reported whether the leaves were dried or not, the amounts used for the extractions are not always given.
It is not reported that validated methods of analysis were used. Not sufficient information is given to repeat the experiment.

Validity of the findings

It is a good point that the authors used microscopic techniques to see what happens within the plants. However, no information is given on the number and/or condition of the chloroplasts, which are the starting point of SG biosynthesis.

In some figures, some outlyers are visible, probably due to sampling methods and/or lack of repetitions of the experiment. (e.g., Fig; 3, 500 µM with colonisation; Fig. 7: A control.

Additional comments

The authors studied the colonisation of mycorrhiza in Stevia. This part of the work seems well-done.
PreIn the introduction, original research papers should be cited, e.g., the first report of the synthesis of steviol by the MEP pathway is by Totté et al. (2000) Tetrahedronn lett. 41, 6407-6410 (not: Tetali 2019). The mention of SGs in Stevia is not from Wölwer-Rieck (2012), etc...
1) Stevia is not an easy plant to work with.
2) The experiment is done only once, and per treatment 6 plants were used. No repetition of the experiments has been done. Therefore, the results reported should be considered as preliminary as more experiments should be done.
3) The number and size of the leaves are not given.
4) Were the leaves used as such or dried before the different extractions? How exactly was the leaf sampling done for the measurements in the different assays? How much material was used?
5) line 182: why is the amount of P and Mg given as a percentage, and not just µg/g fresh weight?
6) line 186: how was the leaf tissue sampled for the pigment analysis? Dry or fresh wt?
7) Extraction of SGs: What is the quantity of leaves extracted (dry or fresh weigth)? Were 3 extractions done on 3 different (weighed) amounts? As shown by Ceunen & Geuns, Plant Science (2013), 198, 72-82, the photoperiod and leaf position on the plants is very important for the SGs content. To obtain the best results, it is advised to dry a huge number of leaves before and grind them to a fine powder. Thereafter, at least 20 mg dry powder should be extracted to avoid extracting a leaf coming from just 1 position of the plant. This should be done at least 3 times.
Was the method for SG analysis validated by the authors?
8) Figure 3 (growth) and Fig. 7 (ST and RebA). The authors did not check the number of chloroplasts in the leaves treated with 200 µM phosphate. Probably, the number did not change after 30 d treatment. The first steps of SG biosynthesis are dependent upon the chloroplasts. Therefore, results should be carefully analysed. The growth increase by 200 µM of about 70 %, and the decrease of Reb A content is about 50 %. Is it possible that the decrease just reflects the “dilution effect” of rebA concentration by the larger cell volume? Of course, for ST this is not the case, but the value for the control is extremely low. Is this due to the one-time experiment, the more as the ST content of the 200 µM treatment with colonisation, as well as of both 1000 µM treatment are not different? A similar problem might exist in Fig. 4 with an “outlyer” of Mg at 500 µM with colonisation.

---

## Round 0.2 · Minor Revisions

Dear Dr. Sarmiento López and Dr. Rodríguez Monroy,

Please find below the comments of the independent reviewers. Although both reviewers are satisfied with the new version and the MS is acceptable for publication, there are still minor editing issues that should be seen to with care.

Looking forward to receiving the revised version,

Ana I. Ribeiro-Barros

Reviewer 1 ·

Basic reporting

See below

Experimental design

See below

Validity of the findings

See below

Additional comments

New report on “Photosynthetic performance and stevioside concentration are improved by the arbuscular mycorrhizal symbiosis in Stevia rebaudiana under different phosphate concentrations”

The authors have taken on board most of my suggestions, and I commend them for improving the manuscript.
A few minor issues to be attended to during editing:
There are still instances of content where concentration should be used, on line 109, 196, 386, 392 and 563.
Line 154 to read ..only one of them are shown.
Line 174 to read ..recorded. To minimize the..
Line 177 do you mean ..positioned on each side of all the nodes of the..
Line 302 mispelled described
Line 315 no comma after data
Line 316 to read .. were checked for normal distributions..
Line 376 to read Leaf number and foliar area were..
Line 377 to read fewer leaves than M+,
Line 378 significantly lower than what..??
I have checked response to the other reviewers’ comments and find them to be in order, but prefer the other reviewers to sign off on them.

Reviewer 2 ·

Basic reporting

The revised manuscript is much better than the original one.

Experimental design

More information is given which makes it better understandable.

Validity of the findings

OK, no problems.

Additional comments

No further comments

---

## Round 0.3 · accepted · Accept

Dear Dr. Sarmiento López and Dr. Rodríguez Monroy,

Thank you for sending the revised version of the MS incorporating the minor revisions suggested by the two independent reviewers. It is my pleasure to inform you that the paper is now accepted for publication.

We acknowledge the choice and efforts to keep the quality standards of PeerJ.

Sincerely

Ana I. Ribeiro-Barros